# Sensitivity of ice loss to uncertainty in flow law parameters in an idealized one-dimensional geometry

Maria Zeitz[1,2], Anders Levermann[1,2,3], and Ricarda Winkelmann[1,2]

[1]Potsdam Institute for Climate Impact Research (PIK), Member of the Leibniz Association, P.O. Box 60 12 03, 14412 Potsdam, Germany
[2]University of Potsdam, Institute of Physics and Astronomy, Karl-Liebknecht-Str. 24-25, 14476 Potsdam, Germany
[3]LDEO, Columbia University, New York, USA.

**Correspondence:** Maria Zeitz (maria.zeitz@pik-potsdam.de), Ricarda Winkelmann (ricarda.winkelmann@pik-potsdam.de)

**Abstract.** Acceleration of the flow of ice drives mass losses in both the Antarctic and the Greenland Ice Sheet. The projections of possible future sea-level rise rely on numerical ice-sheet models, which solve the physics of ice flow, melt, and calving. While major advancements have been made by the ice-sheet modeling community in addressing several of the related uncertainties, the flow law, which is at the center of most process-based ice-sheet models, is not in the focus of the current scientific debate. However, recent studies show that the flow law parameters are highly uncertain and might be different from the widely accepted standard values. Here, we use an idealized flowline setup to investigate how these uncertainties in the flow law translate into uncertainties in flow-driven mass loss. In order to disentangle the effect of future warming on the ice flow from other effects, we perform a suite of experiments with the Parallel Ice Sheet Model (PISM), deliberately excluding changes in the surface mass balance. We find that changes in the flow parameters within the observed range can lead up to a doubling of the flow-driven mass loss within the first centuries of warming, compared to standard parameters. The spread of ice loss due to the uncertainty in flow parameters is of the same order of magnitude as the increase in mass loss due to surface warming. While this study focuses on an idealized flowline geometry, it is likely that this uncertainty carries over to realistic three-dimensional simulations of Greenland and Antarctica.

*Copyright statement.* TEXT

## 1 Introduction

Current and future sea-level rise is one of the most iconic impacts of a warming climate and affects shorelines worldwide (Hinkel et al., 2014; Strauss et al., 2015). The contribution of the large ice sheets in Greenland and Antarctica to sea-level rise sums up to 13.7 + 14.0 mm over the last four decades (Mouginot et al., 2019; Rignot et al., 2019). It has been accelerating in recent years and is expected to further increase with sustained warming (Levermann et al., 2014, 2020; Mengel et al., 2016; Seroussi et al., 2020; Goelzer et al., 2020; Aschwanden et al., 2019; Bamber et al., 2019). Although some convergence can be observed in the projections of the median contribution of ice loss from Antarctica and Greenland, large uncertainties remain,

and coastal protection cannot rely on the median estimate since there is a 50% likelihood that it will be exceeded. Rather, an estimate of the upper uncertainty range is crucial. The most recent IPCC Special Report on the Ocean and Cryosphere in a Changing Climate provides projections of sea-level rise for the year 2100 of $0.43\,\mathrm{m}$ $(0.29 - 0.59\,\mathrm{m})$ and $0.84\,\mathrm{m}$ $(0.61 - 1.10\,\mathrm{m})$ for RCP2.6 and RCP8.5 scenarios, respectively (Pörtner et al., 2019). Other studies find slightly different (Goelzer et al., 2011, 2016; Huybrechts et al., 2011) and partly wider ranges (Levermann et al., 2020). Such projections are typically performed with process-based ice-sheet models which represent the physics in the interior and the processes at the boundaries of the ice-sheet.

In contrast to these processes at the boundaries of the ice sheet, many rheological parameters of the ice are typically not represented as an uncertainty in sea-level projections. The theoretical basis of ice flow, as implemented in ice-sheet models, has been studied in the lab and by field observations for more than half a century and is perceived as well established (Glen, 1958; Paterson and Budd, 1982; Budd and Jacka, 1989; Greve and Blatter, 2009; Cuffey and Paterson, 2010; Schulson and Duval, 2009; Duval et al., 2010). *Glen's flow law*, which relates stress and strain rate in a power law, is most widely used in ice-flow models. It is described in more detail in section 2.1. Some alternatives to the mathematical form of the flow law have been proposed: multi-term power laws like the *Goldsby-Kohlstedt* law or similar (Peltier et al., 2000; Pettit and Waddington, 2003; Ma et al., 2010; Quiquet et al., 2018) and anisotropic flow laws (Ma et al., 2010; Gagliardini et al., 2013) might be better suited to describe ice flow over a wide range of stress regimes. However, they have not been picked up by the ice-modeling community widely, possibly because this would require introducing another set of parameters which are not very well constrained.

Of all flow parameters, the enhancement factor is varied most routinely and its influence on ice dynamics is well understood (Quiquet et al., 2018; Ritz et al., 1997; Aschwanden et al., 2016). However, recent developments suggest that also the other parameters of the flow law are less certain then typically acknowledged in modelling approaches: A review of the original literature on experiments and field observations shows a large spread in the flow exponent $n$ (which describes the nonlinear response in deformation rate to a given stress), which can be between 2 and 4, and the activation energies $Q$ in the Arrhenius law (which describe the dependence of the deformation rate on temperature), which also can vary by a factor of two (Jellinek and Brill, 1956; Butkovich and Landauer, 1960; Mellor and Smith, 1967; Weertman, 1968; Mellor and Testa, 1969; Muguruma, 1969; Barnes et al., 1971; Weertman, 1973; Steinemann, 1958; Bowden and Tabor, 1964; Mellor, 1959; Paterson, 1977; Fletcher, 1970; Goldsby and Kohlstedt, 1997; Craw et al., 2018; Glen, 1952, 1953, 1955; Jellinek and Brill, 1956; Butkovich and J.k, 1957; Raraty and Tabor, 1958; Glen, 1958; Butkovich and Landauer, 1960; Mellor and Smith, 1967; Bromer and Kingery, 1968; Barnes et al., 1971; Steinemann, 1958; Mellor and Testa, 1969; Duval, 1974, 1976; Qi et al., 2017; Treverrow et al., 2012; Duval and Gac, 1982; Nye, 1953, 1957; Gow, 1963; Paterson, 1962; Paterson and Savage, 1963; Holdsworth and Bull, 1970; Hansen and Landauer, 1958; Thomas, 1973; Raymond, 1973; Hobbs, 1974; Paterson, 1977; Martin and Sanderson, 1980; Doake and Wolff, 1985; Blinov and Dmitriev, 1987; Naruse et al., 1988; Talalay and Hooke, 2007; Cuffey and Paterson, 2010; Craw et al., 2018; Qi et al., 2017; Cyprych et al., 2016; Treverrow et al., 2012)(Zeitz et al. submitted). New experimental approaches suggest a flow exponent larger than $n = 3$, which has been the most accepted value so far (Qi et al., 2017). Further, via an analysis of the thickness, surface slope and velocities of the Greenland Ice Sheet from remote sensing data, Bons et al. (2018) relate the driving stress to the ice velocities in regions where sliding is negligible, and can thus infer a flow exponent $n = 4$ under more realistic conditions.

Here we assess the implications of this uncertainty in simulations with the thermomechanically coupled Parallel Ice Sheet Model (the PISM authors, 2018; Bueler and Brown, 2009; Winkelmann et al., 2011), showing that variations in flow parameters have an important influence on flow-driven ice loss in an idealized flowline scenario.

This paper is structured as follows: in Section 2 we recapitulate the theoretical background of ice flow physics and describe the simulation methods used. The results of the equilibrium and warming experiments in a flow-line setup with different flow parameters are presented in Section 3. Section 4 discusses the results and the limitations of the experimental approach and draws conclusions and suggests possible implications of these results.

## 2 Methods

### 2.1 Theoretical background of ice flow physics

The flow of ice cannot be described by the equations of fluid dynamics alone, but needs to be complemented by a material-dependent constitutive equation which relates the internal forces (stress) to the deformation rate (strain rate). Numerous laboratory experiments and field measurements show that the ice deformation rate responds to stress in a nonlinear way. Under the assumptions of isotropy, incompressibility and uni-axial stress this observation is reflected in Glen's flow law, which gives the constitutive equation for ice,

$$\dot{\epsilon} = A\tau^n, \tag{1}$$

where $\dot{\epsilon}$ is the strain rate, $\tau$ the dominant shear stress, $n$ the flow exponent and $A$ the softness of ice (Glen, 1958).

Both, the flow exponent and the softness are important parameters which determine the flow of ice. Usually, the exponent $n$ is assumed to be constant through space and time. Until today, there is no comprehensive understanding of all the physical processes determining the softness $A$. It may depend, among others, on water content, impurities, grain size and anisotropy as well as temperature of the ice. Within the scope of ice-sheet modeling $A$ is typically expressed as a function of temperature alone

$$A = A_0 \exp\left(-\frac{Q}{RT'}\right), \tag{2}$$

where $A_0$ is a constant factor, $Q$ is an activation energy, $R$ is the universal gas constant and $T'$ the temperature difference to the pressure melting point (Greve and Blatter, 2009; Cuffey and Paterson, 2010).

Due to pre-melt processes the softness responds more strongly to warming at temperatures close to the pressure melting point, which is often described by a piece-wise adaption of the activation energy $Q$ (Barnes et al., 1971; Paterson, 1991), with a larger value of $Q$ at temperatures $T' > -10°C$. When using these piece-wise defined values for $Q$ for warm and for cold ice in the functional form of the flow law, the respective factors $A_0$ ensure that the function is continuous at $T' = -10°C$. $A_0$ is therefore dependent on the values of the flow exponent $n$ and both values of $Q$ for cold and for warm ice.

The scalar form of Glen's flow law (Equation (1)) is only valid for uni-axial stresses, acting in only one direction. For a complete picture the stress is described as a tensor of order two. The generalized flow law reads

$$\dot{\epsilon}_{jk} = A(T')\tau_e^{n-1}\tau_{jk}, \tag{3}$$

where $\dot{\epsilon}_{jk}$ are the components of the strain rate tensor and $\tau_{jk}$ are the components of the stress deviator, $\tau_e$ is the effective stress, which is closely related to the second invariant of the deviatoric stress tensor:

$$\tau_e^2 = \frac{1}{2}\left[\tau_{xx}^2 + \tau_{yy}^2 + \tau_{zz}^2\right] + \tau_{xy}^2 + \tau_{xz}^2 + \tau_{yz}^2. \tag{4}$$

Each component of the strain rate tensor depends on all the components of the deviatoric stress tensor through the effective stress $\tau_e$.

Glen's flow law (3) and the softness parametrization (2) are at the center of most numerical ice-sheet and glacier models, independently of the other approximations they might use (the PISM authors, 2018; Winkelmann et al., 2011; Greve, 1997; Pattyn, 2017; Larour et al., 2012; de Boer et al., 2013; Fürst et al., 2011; Lipscomb et al., 2018).

## 2.2 Ice flow model PISM

The simulations in this study were performed with the Parallel Ice Sheet Model (PISM) release stable v1.1. PISM uses shallow approximations for the discretized, physical equations: The shallow-ice approximation (SIA) (Hutter, 1983) and the shallow-shelf approximation (SSA) (Weis et al., 1999) are solved in parallel within the entire simulation domain. The shallow ice approximation is typically dominant in regions with high bottom friction, such that the vertical shear stresses dominate over horizontal shear stresses and longitudinal stresses. The shallow shelf approximation is typically dominant for ice shelves, with zero traction at the base of the ice, and for the fast flow regime in ice streams (Winkelmann et al., 2011). PISM assumes a non-sliding SIA flow and uses the results of the SSA approximations for fast flowing and sliding ice. In PISM, the flow law enters both the SIA and the SSA part of the velocities, as detailed in Winkelmann et al. (2011). It is possible to choose different flow exponents $n$ for the SSA and the SIA, but the softness is the same for both approximations.

The simulations performed here use mostly the SIA mode: The geometry of a two dimensional ice sheet sitting on a flat bed and the SIA mode serve to study the effects of changes in flow parameters onto internal deformation and to separate those effects from changes in sliding, etc. Including the shallow shelf approximation reproduces and even enhances the effect of changes in the activation energies $Q$ (see section 3.5).

## 2.3 Uncertainty in flow exponent and activation energies

The flow exponent $n$ and the activation energies for warm and for cold ice, $Q_w$ and $Q_c$, determine the deformation of the ice as a response to stress or temperature. A recent review (Zeitz et al. submitted, see also literature in the introduction above) reveals a broad range of potential flow parameters $n$, $Q_w$ and $Q_c$. The activation energy for cold ice $Q_c$ is varied between $42\,\mathrm{kJ/mol}$ to $85\,\mathrm{kJ/mol}$ (the typical standard value is $Q_c = 60\,\mathrm{kJ/mol}$). The activation energy for warm ice $Q_w$ is varied between $120\,\mathrm{kJ/mol}$ to $200\,\mathrm{kJ/mol}$ (the standard value is $Q_w = 139\,\mathrm{kJ/mol}$). For the flow exponent $n$, values as low as 1 have been reported, but

since many experiments and observations confirm a nonlinear flow of ice, $n$ has been varied between $2$ and $4$, with the standard value of $n = 3$. The standard values serve as a reference point and correspond to the default values in many ice-sheet models (the PISM authors, 2018; Greve, 1997; Pattyn, 2017; Larour et al., 2012; de Boer et al., 2013; Fürst et al., 2011; Lipscomb et al., 2018).

## 2.4 Adaption of the flow factor $A_0$

The flow factor $A_0$ in the flow law must be adapted to fulfill the following conditions: First, the continuity of the piece-wise defined softness $A(T')$ must be ensured for all combinations of $Q_\mathrm{w}$, $Q_\mathrm{c}$ and $n$. Secondly, a reference deformation rate $\dot{\epsilon}$ at a reference driving stress $\tau_0$ and a reference temperature $T_0'$ (the PISM authors, 2018) should be maintained regardless of the parameters. This is because the coefficient and the power are non-trivially linked when a power law is fitted to experimental data. These conditions give:

$$A_{0,\mathrm{old}} \cdot \exp\left(-\frac{Q_\mathrm{old}}{RT_0'}\right) \cdot \tau_0^{n_\mathrm{old}} = A_{0,\mathrm{new}} \cdot \exp\left(-\frac{Q_\mathrm{new}}{RT_0'}\right) \cdot \tau_0^{n_\mathrm{new}}, \tag{5}$$

$$A_{0,\mathrm{new}} = A_{0,\mathrm{old}} \cdot \exp\left(-\frac{Q_\mathrm{old} - Q_\mathrm{new}}{RT_0'}\right) \cdot \tau_0^{n_\mathrm{old} - n_\mathrm{new}}. \tag{6}$$

If the reference temperature is $T_0' < -10°\mathrm{C}$ the values for cold ice $A_{0,c}$ and $Q_\mathrm{c}$ are used in the equation above, or else $A_{0,w}$ and $Q_\mathrm{w}$ are used. The corresponding $A_{0,\mathrm{new}}$ for warm or cold ice respectively is calculated from the continuity condition at $T' = -10°\mathrm{C}$. For e.g. $T_0' < -10°\mathrm{C}$ it follows

$$A_{0,c,\mathrm{new}} = A_{0,c,\mathrm{old}} \cdot \exp\left(-\frac{Q_{c,\mathrm{old}} - Q_{c,\mathrm{new}}}{RT_0'}\right) \cdot \tau_0^{n_\mathrm{old} - n_\mathrm{new}} \quad \text{and} \tag{7}$$

$$A_{0,w,\mathrm{new}} = A_{0,c,\mathrm{new}} \cdot \exp\left(-\frac{(Q_{c,\mathrm{new}} - Q_{w,\mathrm{new}})}{R \cdot 263.15\mathrm{K}}\right). \tag{8}$$

Here we choose $\tau_0 = 80\mathrm{kPa}$ as a typical stress in a glacier and $T_0' = -20°\mathrm{C}$. Choosing another $\tau_0$ in the same order of magnitude has only little effect on the differences in dynamic ice loss. Choosing another $T_0'$ on the other hand influences how the softness changes with the activation energy $Q$, see Supplemental Figure S1. With $T_0'$ closer to the melting temperature, the difference in softness at the pressure melting point decreases thus the ice loss is less sensitive to changes in the activation energy $Q$.

## 2.5 Experimental design

The study is performed in a flow-line setup, similar to Pattyn et al. (2012), where the computational domain has an extent of 1000km in $x$-direction and 3km in $y$-direction (with a periodic boundary condition). The spatial horizontal resolution is 1 km. The ice rests on a flat bed of length L = 900 km with a fixed calving front at the edge of the bed, such that no ice shelves can form (Figure 1). In contrast to Pattyn et al. (2012), the temperature and the enthalpy of ice sheet are allowed to evolve freely.

The model is initialized with a spatially constant ice thickness and is run into equilibrium for different combinations of flow parameters $Q_\mathrm{c}, Q_\mathrm{w}$ and $n$. The ice surface temperature is altitude dependent, $T_s = -6°\mathrm{C/km} \cdot z - 2°\mathrm{C}$, where $z$ is the surface

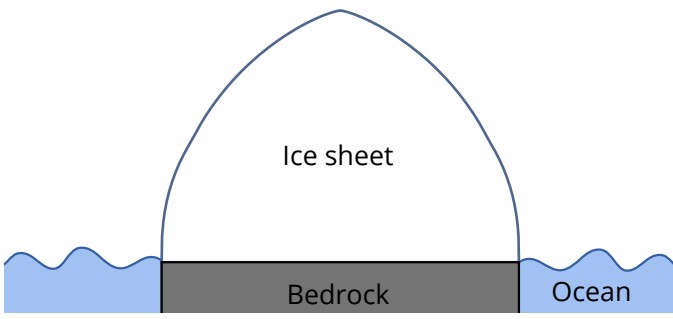

**Figure 1. Sketch of the flow-line setup** The ice is sitting on a flat bed, the fixed calving front does not allow ice-shelves. The accumulation rate is constant throughout the simulation domain and the temperature is altitude dependent.

elevation in km. The accumulation rate is constant in space and time for each simulation. In the warming experiments, for each ensemble member an instantaneous temperature increase of $\Delta T \in [1, 2, 3, 4, 5, 6]°C$ is applied to the ice surface for the duration of 15,000 years (until a new equilibrium is reached), while the climatic mass balance remains unchanged. That means, the temperature increase can lead to an acceleration of ice flow, but is prohibited from inducing additional melt. This idealized
forcing allows us to disentangle the effect of warming on the ice flow from climatic drivers of ice loss.

The thickness profile of the equilibrium state is similar to the *Vialov profile* (see e.g. Cuffey and Paterson (2010); Greve and Blatter (2009)). However, in contrast to the isothermal Vialov profile, here the temperature of the ice is allowed to evolve freely, leading to a non-uniform softness of the ice (the PISM authors, 2018). The extent in $x$-direction is given by the geometry of the setup, a flat bed with a calving boundary condition at the margin, and the height and shape of the ice sheet depend on the
flow parameters $n$, $Q_{\mathrm{w}}$ and $Q_{\mathrm{c}}$ and the accumulation rate $a$.

## 3   Results

### 3.1   Effect of activation energies in model simulations compared to analytical solution

In order to gain a deeper understanding of the influences of $Q_{\mathrm{c}}$ and $Q_{\mathrm{w}}$ on the equilibrium shape of ice-sheets, we here compare the simulated results to analytical considerations based on the Vialov profile.
At a fixed accumulation rate of $a = 0.5\mathrm{m/yr}$, each flow parameter combination leads to an equilibrium state with a thickness profile similar to the Vialov profile but differences in maximal thickness and volume (Figure 2 a). Overall, high activation energies increase ice-flow velocities and reduce the ice-sheet volume. The activation energy for warm ice, $Q_{\mathrm{w}}$, affects the volume and the velocities more strongly than the activation energy for cold ice, $Q_{\mathrm{c}}$. A high $Q_{\mathrm{w}}$ leads to softer ice close to the pressure melting point (supplemental Figure 1) and at the base of the ice sheet, which leads to higher velocities and a lower
equilibrium volume of the ice sheet while a low $Q_{\mathrm{w}}$ leads to stiffer ice close to the pressure melting point and at the base of the

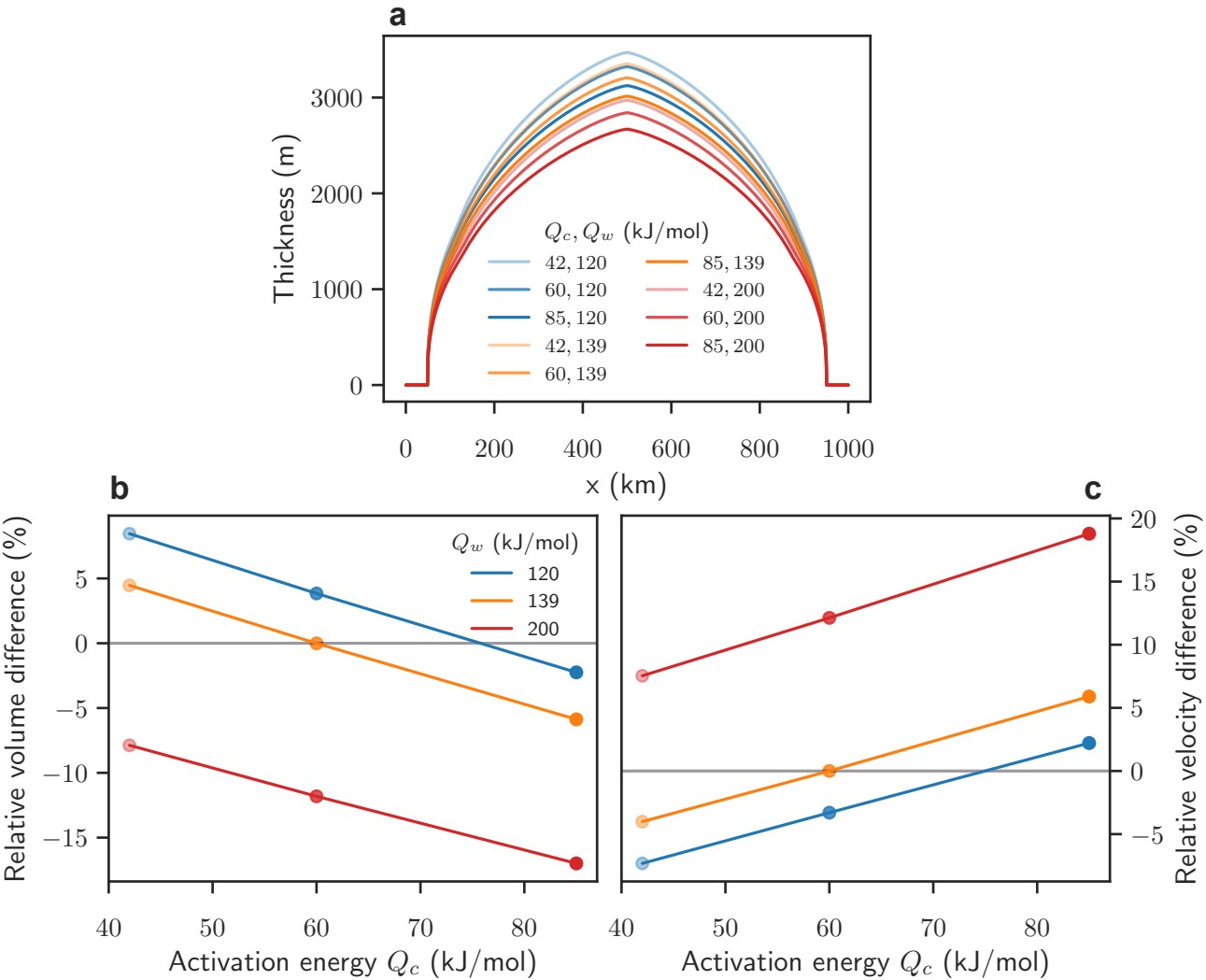

**Figure 2. Effect of activation energies on equilibrium volume and velocities** with fixed accumulation rate $a = 0.5\,\text{m/yr}$ and flow exponent $n = 3$. Thickness profiles of equilibrium states for different combinations of activation energies $Q_w$ and $Q_c$ **(a)**. Relative difference of average equilibrium volumes **(b)** and velocities **(c)** compared to the reference state with standard parameters for parameter combinations of $Q_w$ and $Q_c$. $Q_c$ is shown on the x-axis and $Q_w$ is given through the color of the the markers (Blue: $Q_w = 120\,\text{kJ/mol}$, orange: $Q_w = 139\,\text{kJ/mol}$, red: Blue: $Q_w = 200\,\text{kJ/mol}$)

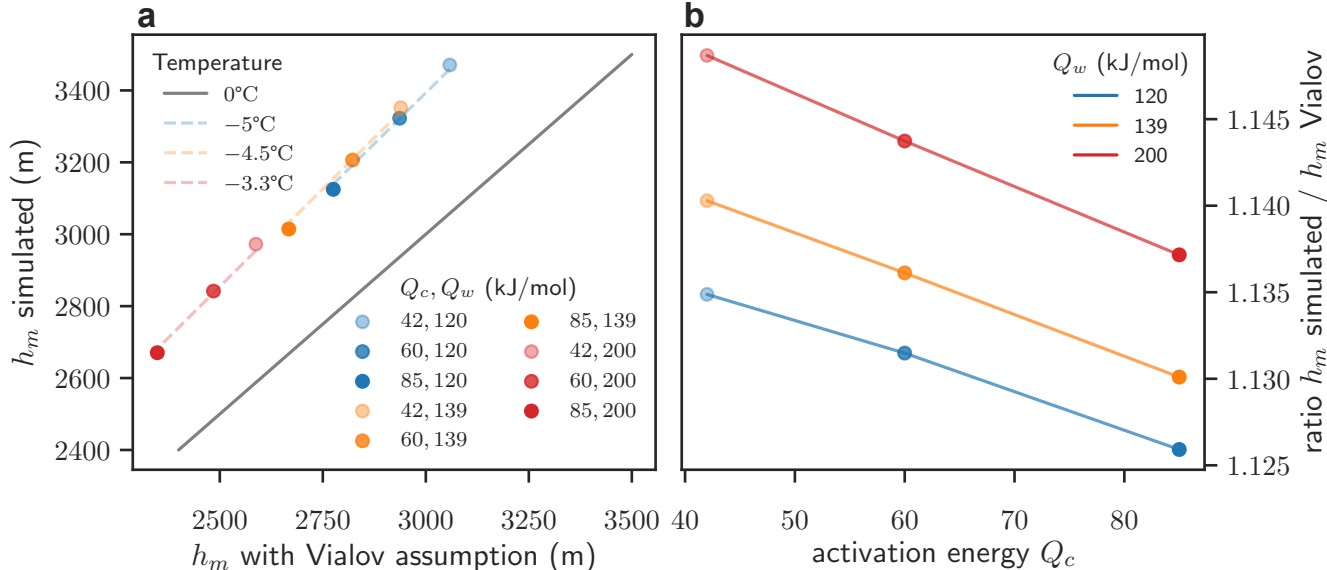

**Figure 3. Comparison of simulated equilibrium thickness with analytical results: (a)** Dots: Maximal thickness $h_m$ of the simulated polythermal ice sheet versus the analytical solution for the maximal thickness $h_m$ of a temperate Vialov profile with the same flow parameters and accumulation rate. Colors indicate the flow-parameter combination. The grey line indicates identity. Short, dashed lines indicate the analytical $h_m$ with a temperature lower than the pressure melting point versus $h_m$ at the pressure melting point with the same flow parameters and accumulation rate. The temperature, which fits the simulated results best, is indicated in the legend. **(b)** Ratio of the simulated $h_m$ to the analytic $h_m$ (assuming a temperate ice sheet) versus $Q_c$ for different parameter combinations $Q_c, Q_w$. The value of $Q_w$ is indicated by the color.

ice sheet and in consequence the velocities decrease and the volume increases (Figure 2, b and c). For a fixed $Q_w$, the volume appears to decrease linearly with increasing $Q_c$ and the velocity appears to increase linearly with increasing $Q_c$.

The maximal thickness of an isothermal ice sheet can be estimated with the Vialov profile

$$h_m = 2^{n/(2n+2)} L^{1/2} \left( \frac{(n+2)a}{2A(T')(\rho g)^n} \right) \tag{9}$$

with the Glen exponent $n$, the ice sheet extent $2L$, the pressure adjusted temperature $T'$, the gravity $g$, and the ice density $\rho$ (Greve and Blatter, 2009). The Vialov thickness of a temperate ice sheet (isothermal at the pressure melting point), where the softness is evaluated at the pressure melting point depending on the activation energies $Q_c$ and $Q_w$ (see Equation (2)) gives a lower bound to the thickness, given the same geometry and flow parameters. The simulated maximal thickness is larger than the lower bound for all parameter combinations (Figure 3 a, lower bound indicated by grey line) and the ratio between the

maximal thickness $h_m$ from the PISM simulation to the lower bound from the Vialov profile depends on both, $Q_w$ and $Q_c$. The ratio increases with higher $Q_w$ and decreases with higher $Q_c$ (Figure 3 b). The ice-sheet thickness of the polythermal ice sheet, as simulated with PISM, matches well the Vialov thickness calculated with Equation (9), if an effective temperature

$T'_{\text{eff}} < 0°C$ is assumed. The effective temperature $T'_{\text{eff}}$, which matches simulations best, varies for different $Q_{\text{w}}$. For $Q_{\text{w}} = 120$ kJ/mol, an effective temperature of $T'_{\text{eff}} = -5°C$ matches well the equilibrium thickness of the polythermal ice sheets. For $Q_{\text{w}} = 200$ kJ/mol, an effective temperature of $T'_{\text{eff}} = -3.3°C$ matches well the equilibrium thickness of the polythermal ice sheets. These differences can be partly explained by the altitude-dependent surface temperature: The maximal thickness of the ice sheets varies by approximately $800$ m, which leads to a difference in ice surface temperature of approximately $4.8°C$ between the thickest and the thinnest ice and thus influences the temperature within the ice sheet.

The relative difference of average velocities $d_v = (\bar{v} - \bar{v}_0)/\bar{v}_0$ spans from $d_v = -7\%$ (with a corresponding relative difference in ice sheet volume of $d_{vol} = +10\%$) for the lowest combination of activation energies to $d_v = +18\%$ with a difference in volume of $d_{vol} = -15\%$ for the highest combination of values for $Q_{\text{c}}$ and $Q_{\text{w}}$ (Figure 2 b).

## 3.2 Ice-sheet initial states

In order to keep the initial ice volume largely fixed (with variations of less than one percent) in the warming experiments, we adapt the accumulation rate for each parameter combination of $Q_{\text{c}}$ and $Q_{\text{w}}$.

Since simulations with high activation energies $Q_{\text{w}}$ have a smaller equilibrium volume at the same accumulation rate than simulations with standard activation energies, the accumulation rate $a$ is increased to maintain an equilibrium volume close to the reference value. Simulations with low activation energies $Q_{\text{c}}$ have a higher volume at the same accumulation rate, so the accumulation rate $a$ is decreased. In the case of an isothermal ice sheet the maximal thickness and the volume can be computed analytically as shown above in Equation (9). In our model simulations, however, the temperature distribution within the ice can evolve freely, thus the softness is not uniform and an analytical solution cannot be found.

In order to find the right adaptation for the accumulation rates, we start from the volume calculated from the isothermal approximation as a first guess and run the model into equilibrium. If the relative difference between the new equilibrium volume and the standard equilibrium volume exceeds 1%, we further change the accumulation rate and repeat the equilibrium simulation, always starting from the same initial state. The final equilibrium states found via this iterative approach differ by max. 0.8% in ice volume (supplemental Figure S2) and the difference in maximal thickness is less than 100m (Figure 4, a and b).

For the combination of high activation energies $Q_{\text{w}}$ and $Q_{\text{c}}$, the relative differences $d_x = (x - x_0)/x_0$ of both, adapted accumulation rates $a$ and mean surface velocities $v$, increase by more than 300% (Fig. 4, c and d) and for the combination of low activation energies $Q_{\text{c}}$ and $Q_{\text{w}}$ both, adapted accumulation rates $a$ and surface velocities $v$ are approximately 50% lower compared to the case with standard parameters. Both, the accumulation rate and the velocities, change in the same way since they balance each other in equilibrium.

The maximal thickness of the polythermal simulated ice sheet is approximately 13-16% larger than the lower bound estimated with a temperate ice sheet (Figure 5, a and b) with the same flow parameters and accumulation rates. Similar to the case with fixed accumulation rates, the simulated thickness matches the Vialov thickness well, if an effective temperature $T'_{\text{eff}} < 0°C$ is assumed. The effective temperature, which mathces simulations best, varies for different $Q_{\text{w}}$, from $-5°C$ for $Q_{\text{w}} = 120$ kJ/mol to $-3.6°C$ for $Q_{\text{w}} = 200$ kJ/mol. This difference can not be sufficiently explained by variations in surface

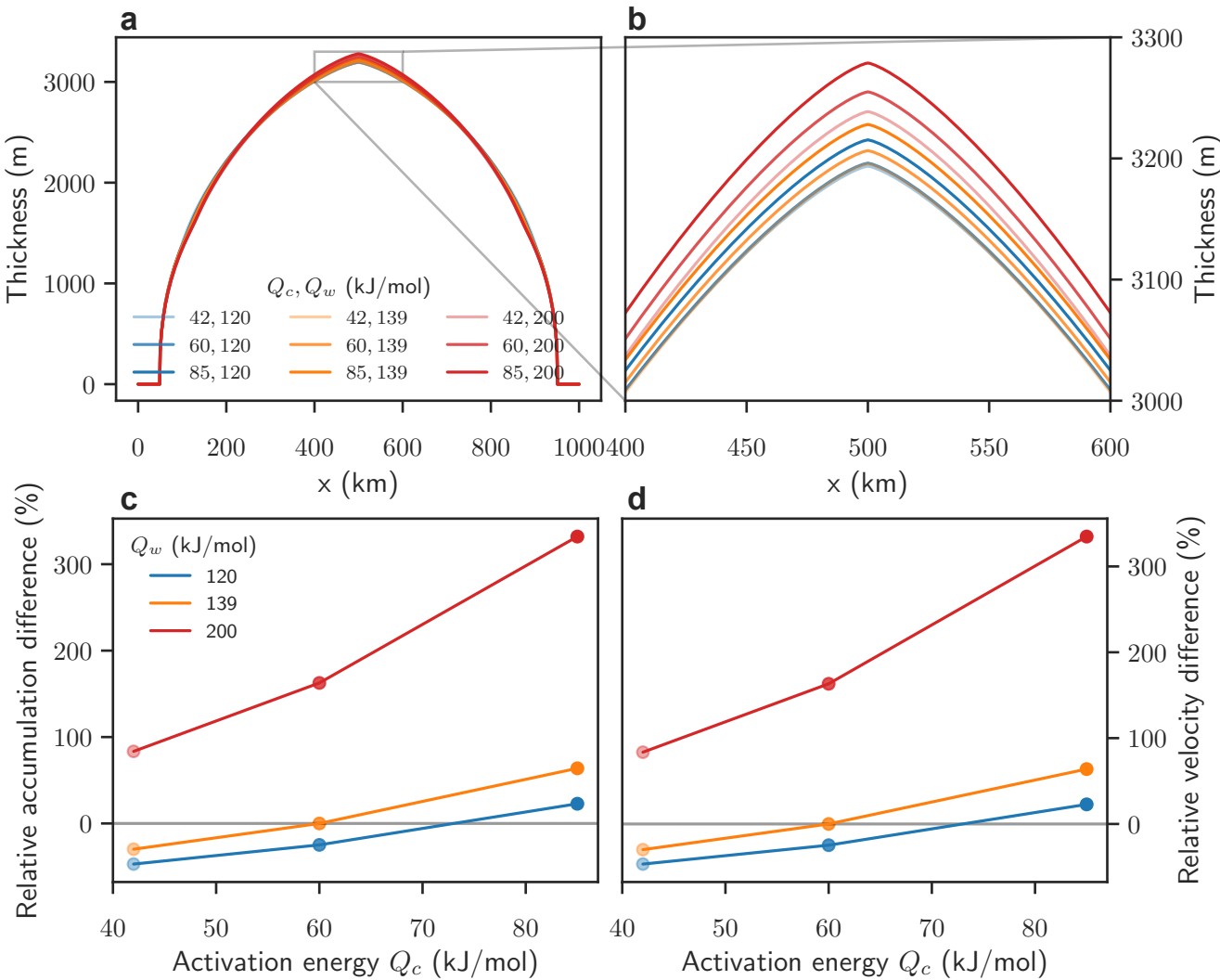

**Figure 4. Effect of flow parameters on equilibrium state without warming with adapted accumulation rates** and flow exponent $n = 3$.
**(a)** Thickness profile of equilibrium states for parameter combinations of $Q_w$ and $Q_c$ with a zoom on the ice divide **(b)**. Relative difference of accumulation rates **(c)** needed to keep the ice sheet volume in equilibrium close to the reference simulations with standard flow parameters and relative difference in average surface velocities **(d)** versus $Q_c$. The value of $Q_w$ is given by the color.

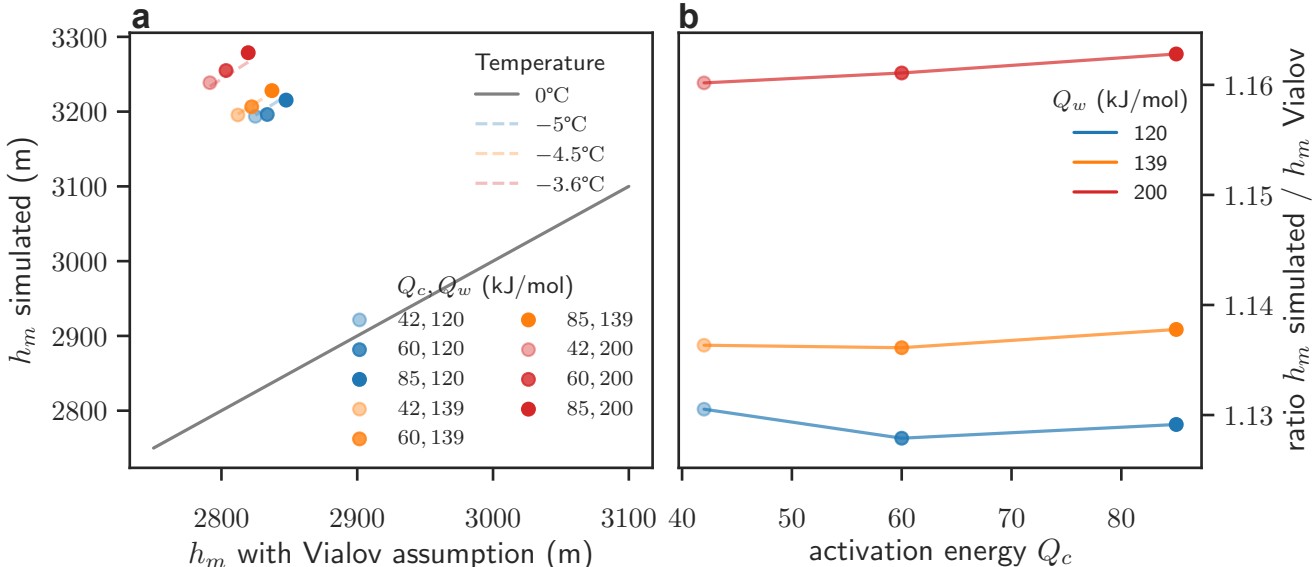

**Figure 5. Comparison of simulated equilibrium thickness with analytical results:** **(a)** Dots: Maximal thickness $h_m$ of the simulated polythermal ice sheet versus the analytical solution for the maximal thickness $h_m$ of a temperate Vialov profile with the same flow parameters and accumulation rate. Colors indicate the parameter combination. The grey line indicates identity. Short, dashed lines indicate the analytical $h_m$ with a temperature lower than the pressure melting point versus $h_m$ at the pressure melting point with the same flow parameters and accumulation rate. The temperature, which fits the simulated results best, is indicated in the legend. **(b)** Ratio of the simulated $h_m$ to the analytic $h_m$ (assuming a temperate ice sheet) versus $Q_c$ for different parameter combinations $Q_c, Q_w$. The value of $Q_w$ is indicated by the color.

temperature due to the difference in ice-sheet thickness. Rather the higher effective temperatures are linked to increased flow velocities of the ice, which in turn might lead to strain heating. In simulations with a high $Q_w$ the simulated thickness has a higher discrepancy to the estimated lower bound (assuming a temperate ice sheet) than simulations with a low $Q_w$. In contrast

to the case with fixed accumulation rate (Figure 3) the ratio between the estimated and the simulated thickness depends only very little on $Q_c$.

### 3.3 Flow-driven ice loss under warming

Disentangling the purely flow-driven ice losses from the influences of melting, different initial temperature profiles and variations in sliding requires several conditions: 1) The initial volume is fixed, which is here attained through adjustment of the

220 accumulation rate for the different flow parameter combinations as explained in section 3.2. 2) The surface mass balance is fixed, i.e., we do not allow for additional melt, and the accumulation rate does not change with warming. 3) Sliding is effectively inhibited (which is here ensured by applying an SIA-only condition).

The effect of the temperature increase is limited to warming at the ice surface which can propagate into the interior of the ice sheet though diffusion and advection. Warming makes the ice softer thus accelerates the flow and ice discharge. Since temperature diffusion in an ice sheet is a very slow process, we apply the temperature anomaly for a total duration of 15,000 years. The total mass balance is evaluated and compared to the standard parameter simulation after 100, 1000 and 10,000 years of warming. A new equilibrium state is reached after 10,000 years for all parameter combinations (see longer time-series in Supplemental Figure S3).

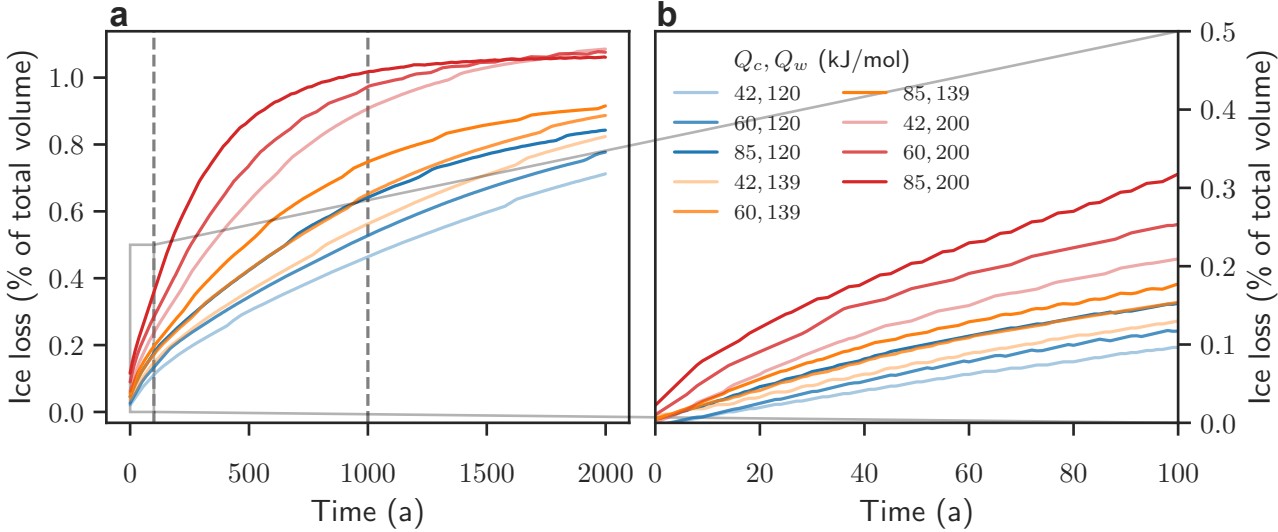

**Figure 6. Time series for flow driven ice discharge under 2°C warming: (a)** Time evolution of ice loss with different activation energies $Q_c$ and $Q_w$ and the flow exponent $n = 3$, subject to a temperature anomaly forcing of $\Delta T = 2°C$. **(b)** Zoom on the first 100 years.

In the experiments, the ice sheet loses mass for all warming levels and all parameter combinations. However, the amount and rate of the ice loss is dependent on the flow parameters. Figure 6 shows the ice-sheet response to a warming of 2°C. For a fixed flow exponent of $n = 3$ the fastest ice loss is observed for the flow parameter combination of $Q_c = 85 \, \text{kJ/mol}$ and $Q_w = 200 \text{kJ/mol}$ and the slowest ice loss for $Q_c = 42 \, \text{kJ/mol}$ and $Q_w = 120 \text{kJ/mol}$. Simulations with $Q_w = 200 \text{kJ/mol}$ reach a new, temperature adapted equilibrium already after 2,000 yrs, while simulations with lower $Q_w$ continue to lose mass.

The sensitivity to variations in flow parameters is measured via the relative differences for flow-driven ice loss $d_m = (\Delta m - \Delta m_0)/\Delta m_0$, where the reference $\Delta m_0$ is always given by the simulation with standard parameters under the same temperature increase (Figure 7). While the long term response to warming, after 10,000 years, is not very sensitive to the particular choice of flow parameters, the rate of flow-driven ice loss is. The largest relative differences in ice loss is found in the first century after the temperature increase (Figure 7, a), indicating that high $Q_w$ speed up the flow-driven ice loss. Under 2°C of warming, ice loss after 100 years is enhanced more than two-fold (i.e. increased by up to 118%) in simulations with $Q_w = 200 \, \text{kJ/mol}$, while low $Q_w$ reduces the relative ice loss by up to 37%.

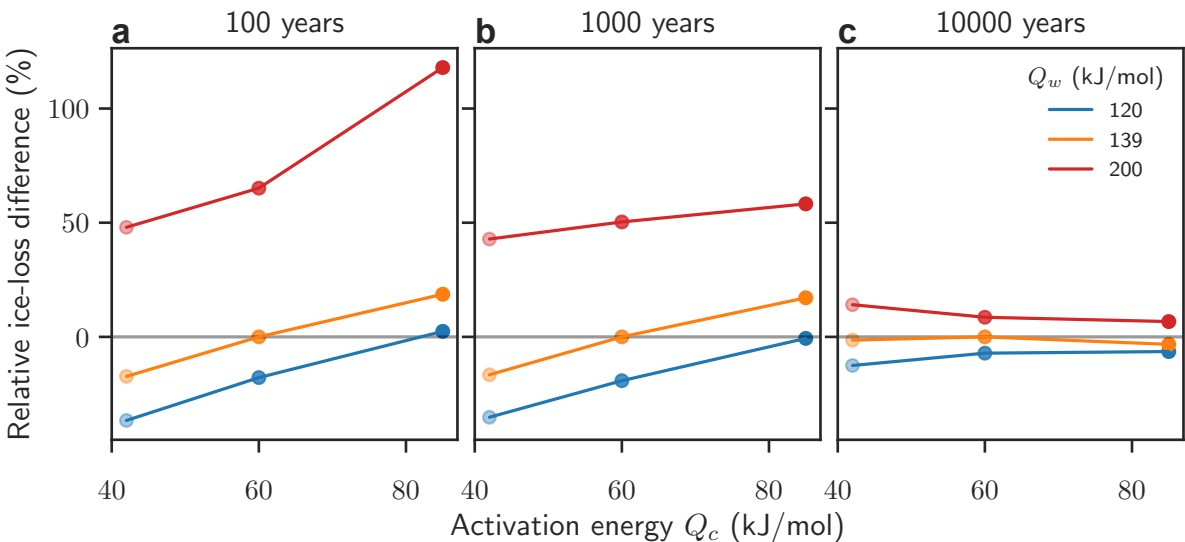

**Figure 7. Effect of activation energy on flow driven ice discharge under 2°C warming:** Relative difference of flow-driven ice loss after 100 **(a)**, 1000 **(b)** and 10,000 **(c)** years versus $Q_c$. The value of $Q_w$ is given by the color. The simulations have reached a new equilibrium after 10,000 years.

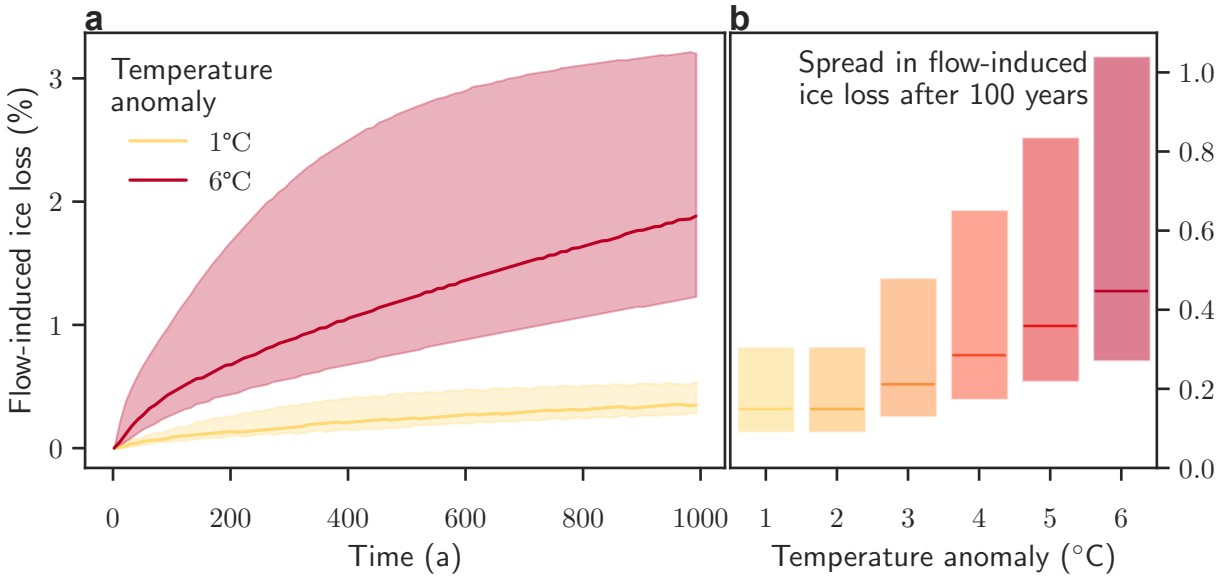

**Figure 8. Effect of temperature forcing and activation energy on flow-driven ice (a)** Time evolution of ice loss under warming of $1\,°C$ and $6\,°C$. For warming of $\Delta T = 1\,°C$ the conceptual ice sheet loses 0.35% of ice after 1000 yr for standard parameters (solid, yellow line). Variations in the activation energies $Q$ lead to variations in ice loss on the same order of magnitude (shaded area). For a warming of $\Delta T = 6\,°C$ the conceptual ice sheet loses 1.89% of ice after 1000 yr for standard parameters (solid, red line). The variations in ice loss due to different parameters for the activation energy $Q$ (shaded area) are strongly asymmetrical and, in particular during the first 300 years, high compared to the total amount of ice loss. **(b)** Uncertainty in flow induced ice loss after 100 years of simulation time over all combinations of $Q_w$, $Q_c$ and temperature anomalies $\Delta T$. The flow exponent $n = 3$ is kept fixed for all simulations.

The effect of the flow parameters on flow-driven ice loss upon warming is robust for different temperature increases. Ice losses as well as the spread in flow-driven ice loss both increase for higher warming levels (see Figure 8). For a warming of $\Delta T = 1\,^{\circ}\mathrm{C}$ the idealized ice sheet loses 0.09% after 100 yr and 0.35% of ice after 1000 yr for standard parameters. For a warming of $\Delta T = 6\,^{\circ}\mathrm{C}$ the ice sheet loses 0.46% after 100 yr and 1.89% of ice after 1000 yr for standard parameters (solid, red line). For comparison, the Greenland Ice Sheet has lost approximately 0.18% of its mass in the period between 1972 and 2018 (Mouginot et al., 2019), which includes all processes: increase in flow, melting, and sliding.

The effect of flow parameter changes onto the purely flow-driven ice loss after 100 years is of the same order of magnitude as the effect of surface warming by several degrees. In particular the uncertainty ranges of ice loss for warming of $2\,^{\circ}\mathrm{C}$ and warming of $6\,^{\circ}\mathrm{C}$ overlap (Figure 8 b), when solely considering the ice loss is driven by changes in flow and excluding surface mass balance changes.

## 3.4 Influence of the flow exponent $n$

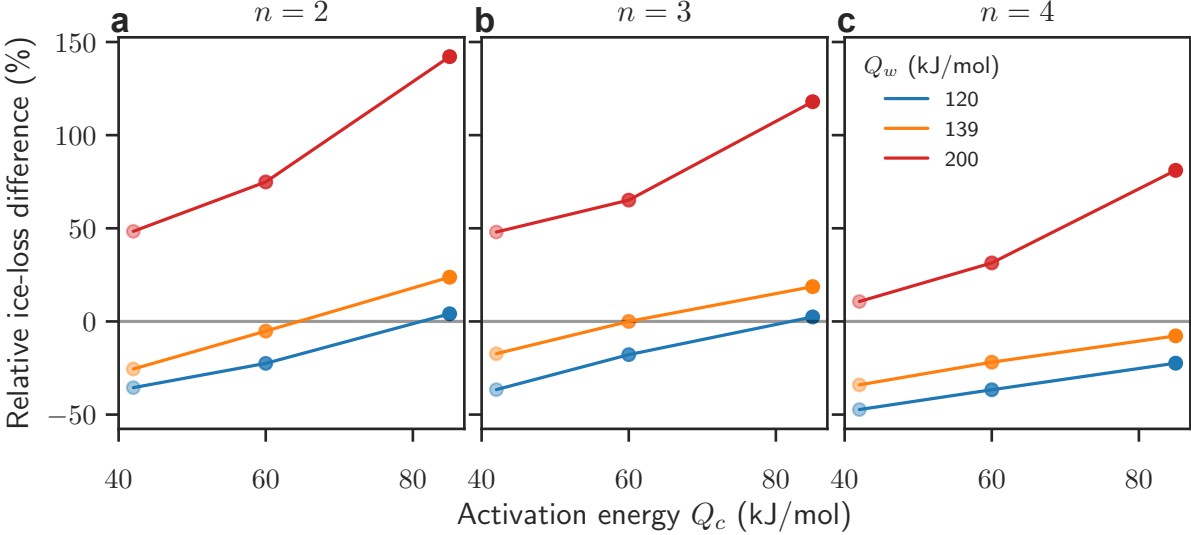

**Figure 9. Effect of the flow exponent and activation energies on flow-driven ice loss after 100 years under $2\,^{\circ}\mathrm{C}$ of warming:** Relative difference in flow-driven ice discharge for $n = 2$ **(a)**, $n = 3$ **(b)** and $n = 4$ **(c)** for different combinations of the flow exponent $n$ and activation energies $Q_\mathrm{c}$ and $Q_\mathrm{w}$. The reference is always a simulation performed with standard parameters $n = 3$, $Q_\mathrm{c} = 60\,\mathrm{kJ/mol}$ and $Q_\mathrm{w} = 139\,\mathrm{kJ/mol}$.

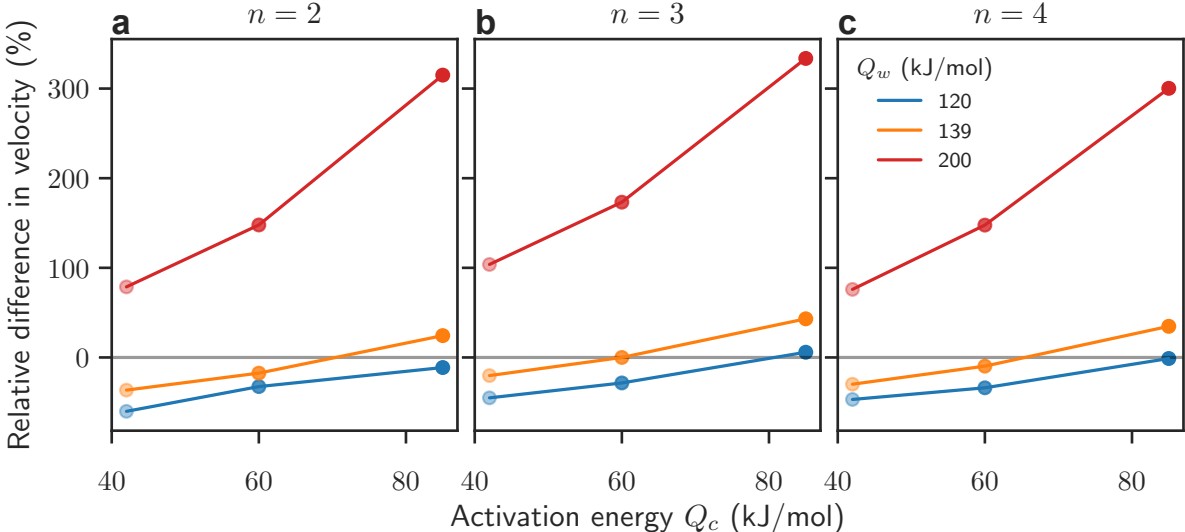

**Figure 10. Effect of the flow exponent and activation energies on mean velocity change after 100 years under $2°$C of warming:** Relative difference in average surface velocity f or $n = 2$ **(a)**, $n = 3$ **(b)** and $n = 4$ **(c)** for different combinations of the flow exponent $n$ and activation energies $Q_c$ and $Q_w$. The reference is always a simulation performed with standard parameters $n = 3$, $Q_c = 60$ kJ/mol and $Q_w = 139$ kJ/mol. Variations in the flow exponent $n$ do not significantly influence the relative difference of mean velocities after 100 years.

Variations in the flow exponent $n$ do not change the qualitative effect of variations in activation energies $Q$ on the ice loss. After 100 years for a temperature anomaly of $\Delta T = 2°$C a higher $n$ seems to mitigate the effect of the activation energy on differences in ice loss, while a lower $n$ seems to enhance this effect (Figure 9). However, the effect of variations in activation

energy on the average surface velocity is almost independent of the choice for the flow exponent $n$ (Figure 10).

The influence of the activation energies $Q_c$ and $Q_w$ on ice flow is similar even with different flow exponents $n$. This is robust for different warming scenarios from $+1$ to $+6°$C. A higher flow exponent $n$, which leads to a more pronounced nonlinearity in ice flow, does not enhance but reduce variations in dynamic ice loss. This might be linked to the details of the parametrization in the flow law: As seen in both equations (6) and (8), the factor $A_0$ decreases if the flow exponent $n$ is

260 reduced and $n_{\text{new}} < n_{\text{old}}$. Compared to the nonlinear stress dependency $\tau^n$ in the flow law the temperature dependent softness $A(T') = A_0 \cdot \exp(-Q/RT')$ becomes less important with increasing flow exponent $n$.

### 3.5 Robustness of results to changes in accumulation and sliding

The overall effect of uncertainties in the activation energies $Q$ remains robust, even if an additional driver of ice loss is taken into account. In a simulation where in addition to warming of $2°$C we also reduce the accumulation rate by 50%, the ice losses

remain dependent on the flow parameters $Q_c$ and $Q_w$ (Figure 11, lines indicate results without a change in accumulation rate, analogous to Figure 7 and squares indicate results with an additional 50% decrease in accumulation rate). After 100 years of forcing the relative spread of ice loss is slightly larger if accumulation changes are included. In particular, for $Q_w = 200$ kJ/mol,

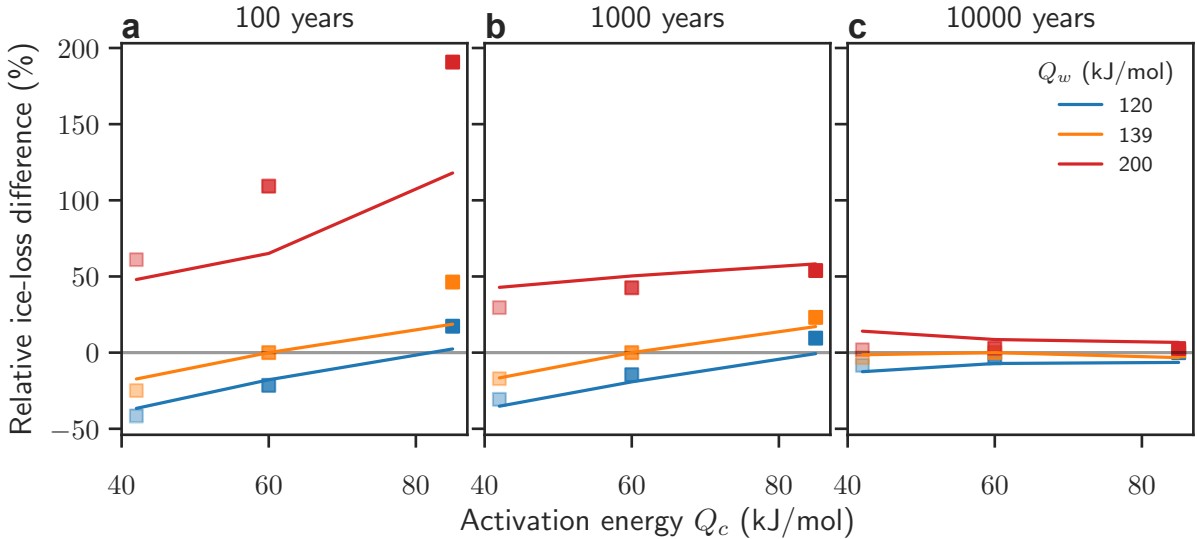

**Figure 11. Effect of the flow exponent and activation energies on flow-driven ice loss under** $2°\text{C}$ **of ice warming in combination with a 50% reduction in accumulation rates:** Relative difference in flow-driven ice discharge after 100 **(a)**, 1000 **(b)** and 10,000 **(c)** years. The ice sheet has reached a new equilibrium after 10,000 years. Relative difference for $2°\text{C}$ warming with an additional reduction of the accumulation rate of 50% (squares) are compared to the results without changes in the accumulation rate (lines, also see Figure 7).

the relative increase of mass loss mounts from 118% to 190%. On longer time scales, the spread in ice loss is reduced (after 10,000 years of forcing, when the ice sheet has reached a new equilibrium, the relative spread is below $\pm10\%$).

When sliding is taken into account via the shallow ice approximation for sliding ice (see the PISM authors (2018)) the uncertainty in flow parameters leads to relative changes in ice loss from -30% to +470% after 100 years, which is a considerably larger spread than without sliding. The relative differences decrease with time, but remain ranger than without sliding. After 1000 years the ensemble member with low activation energies lost 40% less ice than the standard parametization and high activation energies alsmost double the ice loss (+90%). After 10,000 years, when the ice sheets have reached a new equilibrium,

the relative differences still range from -16% to +40% (see Figure 12).

## 4    Discussion and Conclusion

In this study we present a first attempt to disentangle and quantify the effect of uncertainties in the flow law parameters, in particular the activation energies $Q$ and the flow exponent $n$, onto ice dynamics.

The effect of ice rheology in ice-sheet models has been adressed in several studies with different experimental setups and

different time frames. In particular the effect of the enhancement factors, which are often used to approximate the change in ice flow due to anisotropy, has been explored (Ritz et al., 1997; Ma et al., 2010; Humbert et al., 2005; Quiquet et al., 2018). In addition, the effect of the initial conditions (Seroussi et al., 2013; Nias et al., 2016; Humbert et al., 2005) and the effect

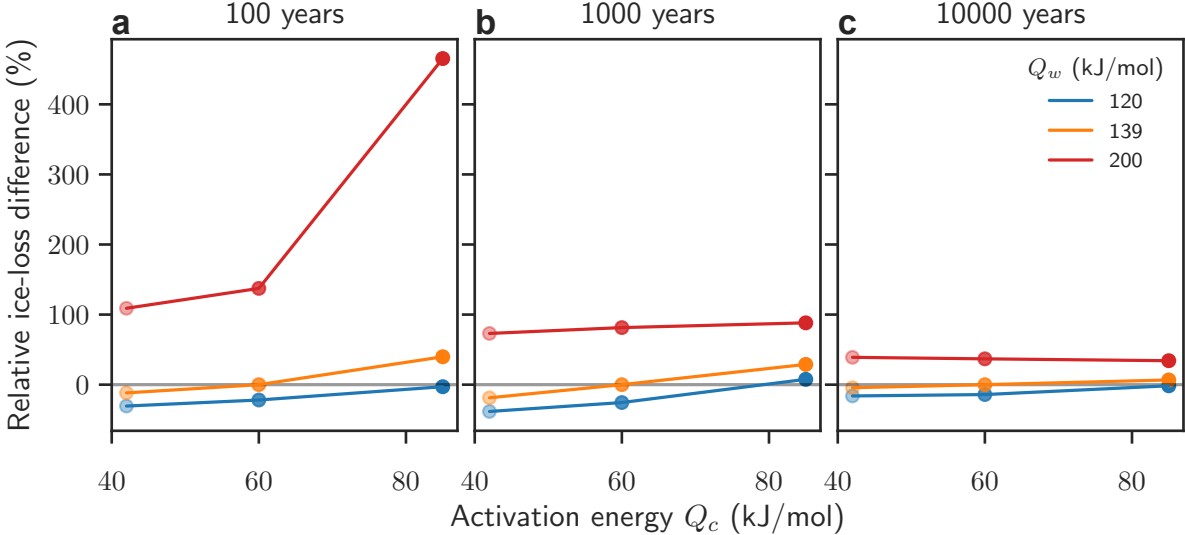

**Figure 12. Effect of activation energy on flow driven ice discharge under 2°C warming, including sliding:** Relative difference of flow and sliding driven ice loss after 100 **(a)**, 1000 **(b)** and 10,000 **(c)** years. The simulations have reached a new equilibrium after 10,000 years.

of the mathematical form of the flow law itself (Quiquet et al., 2018; Peltier et al., 2000; Pettit and Waddington, 2003) have been studied. These studies have been crucial for the understanding of different enhancement factors in the shallow ice and the shallow shelf approximation (Ma et al., 2010), for the reconciliation of the aspect ratios of the Greenland Ice Sheet and the Laurentide Ice Sheet during the last glacial maximum (Peltier et al., 2000) and the ice flow in Antarctica and the Greenland Ice Sheet (Ritz et al., 1997; Seroussi et al., 2013; Quiquet et al., 2018; Nias et al., 2016; Humbert et al., 2005).

However, the approach presented in this manuscript is different in two important aspects: Firstly, the systematic study of not only the flow exponent $n$ but also the activation energies $Q$ has not been performed so far. Secondly, the idealized experimental setup, as presented in this study, allows to disentangle the effects of the flow itself from other drivers and other sources of uncertainty. Several conditions need to hold to this end: The ice sheet is sitting on a flat bed and its maximal extent is determined by a calving front at the borders of the bed, thus no ice-ocean interactions or impacts of the bed geography influence the ice flow. Sliding is generally inhibited (the ice dynamics is described by the shallow ice approximation, with zero basal velocity), no changes in sliding velocity influence the ice flow. The accumulation rate is fixed and independent of the temperature change, so that the ice loss is only driven by changes in flow and not by melting. These idealizations allow a clear understanding of the impact of the flow exponent and the activation energies on ice flow. In addition, they allow to compare the simulations of the polythermal ice sheet to the analytically solvable limit of an isothermal ice sheet by using the Vialov approximation.

In this setup the largest effect of the uncertainties in the flow parameters is observed in the first century after warming, while the effect of the uncertainties on ice loss becomes less important as the ice approaches a new equilibrium. Uncertainties in

the activation energies alone account for up to a doubling in ice loss during the first 100 years of warming and are on the same order of magnitude as the effects of increased temperature forcing, under fixed surface mass balance. This effect remains robust, even if changes in the surface mass balance are taken into account. Reducing the surface mass balance by 50%, which is comparable to the changes in total surface mass balance of the Greenland Ice Sheet from 1972 to 2012 (Mouginot et al., 2019), increases the effect of the flow parameters on a timescale of 100 years and remains comparable on a timescale of 1000 years. Only as the ice sheet approaches its new equilibrium, the effect of the flow parameters becomes negligible. Allowing for not only flow but sliding while keeping all other conditions equal increases the effect of flow parameters substantially, leading to up to a five-fold increase in ice loss after 100 years compared with standard parameters.

Acknowledging the uncertainty in flow parameters might slightly shift the interpretation of previous studies. For instance, the effect of the initial thermal regime, as studied by Seroussi et al. (2013) could be enhanced if the the activation energies were higher than assumed, by making the ice softness more sensitive to changes in temperature. The crossover stress in the multi-term flow law presented by Pettit and Waddington (2003), at which the linear and the cubic term are of the same importance, is highly sensitive to the values of the activation energies. The positive feedback through shear heating, as studied for example by Minchew et al. (2018), could also be enhanced if activation energies were higher than usually assumed. The uncertainty in the flow-law parameters may further provoke a re-evaluation of other parameters, e.g. concerning melting and basal conditions. In particular, Bons et al. (2018) thorough analysis of observational data of the Greenland Ice Sheet supports a flow exponent of $n = 4$, not the standard value of $n = 3$, which is is in line with recent laboratory experiments which also find $n > 3$ (Qi et al., 2017). Assuming a higher flow exponent $n = 4$ has shown to significantly reduce the previously assumed area where sliding is possible (Bons et al., 2018; MacGregor et al., 2016). Moreover, both the flow exponent $n$ and the activation energies $Q$ feed into the grounding line flux formula (Schoof, 2007). In several ice-sheet models, this formula is used to determine the position and the flux over the grounding line in transient simulations (Reese et al., 2018). A change in the flow parameters $n$ and $Q$ has thus implications for the advance and retreat of grounding lines in simulations of the Antarctic Ice Sheet and possibly the onset of the marine ice sheet instability, a particularly relevant process for the long-term stability of the Antarctic Ice Sheet. On the Greenland Ice Sheet increased ice flow might drives ice masses into ablation regions, where the ice melts. A possible effect of uncertainty in flow parameters on this particular feedback remains to be explored. Aschwanden et al. (2019) have found that uncertainty in ice dynamics plays a major role for mass loss uncertainty during the first 100 years of warming. While their study attributes the uncertainty mostly to large uncertainties in basal motion and only to a lesser extent to the flow via the enhancement factor, the uncertainties of the flow law and of the basal motion are not independent, as suggested by e.g Bons et al. (2018).

While the conclusions from the idealized experiments presented here cannot be transferred directly to assess uncertainty in sea-level rise projections, they are an important first step which allows informed choices about parameter variations in more realistic simulations of the Greenland or Antarctic Ice Sheet.

*Code and data availability.* Data and code are available from the authors upon request.

*Author contributions.* R.W. and A.L conceived the study. M.Z., A.L. and R.W. designed the research and contributed to the analysis. M.Z. carried out the literature review and the analysis. M.Z., R.W. and A.L. wrote the manuscript.

*Competing interests.* No competing interest.

*Acknowledgements.* M.Z. and R.W. are supported by the Leibniz Association (project DOMINOES). R.W. is grateful for support by the Deutsche Forschungsgemeinschaft (DFG) through grants WI4556/3-1 and WI4556/5-1 and by the PalMod project (FKZ: 01LP1925D), supported by the German Federal Ministry of Education and Research (BMBF) as a Research for Sustainability initiative (FONA). This research was further supported by the European Union's Horizon 2020 research and innovation programme under grant agreement no. 820575 (TiPACCs). Development of PISM is supported by NASA grant NNX17AG65G and NSF grants PLR-1603799 and PLR-1644277. The authors gratefully acknowledge the European Regional Development Fund (ERDF), the German Federal Ministry of Education and Research and the Land Brandenburg for supporting this project by providing resources on the high performance computer system at the Potsdam Institute for Climate Impact Research. We thank Hilmar Gudmundsson, David Prior and Thomas Kleiner for insightful discussions.

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
