# Peer review of "Sensitivity of ice loss to uncertainty in flow law parameters in an idealized one-dimensional geometry"

_The Cryosphere, 2020_

## Referee Comment (RC1) · Anonymous Referee #1 · 10 May 2020

This paper presents the results of a very creative and important inquiry into the question of how uncertainty in ice flow-law parameters may impact the predictions numerical ice-sheet models make for future sea level rise. The results of the work are very convincing and the case is well made to attend to flow-law uncertainty with greater effort in the future. The work is conducted with a simple numerical model under simple idealized experiments. Thus, the work reaches substantial conclusions that are not impacted by other, extraneous details.

I have put most of my minor editorial comments and questions in the marked-up pdf manuscript that I provide as an attachment to the review.

I think that panel B of figure 1 could be re-drafted either using a log scale (not sure if that would work) or just a focus on the top of the ice sheet, so that all the curves don't

simply plot one on top of another (as is the case now).

Please also note the supplement to this comment:
https://www.the-cryosphere-discuss.net/tc-2020-79/tc-2020-79-RC1-supplement.pdf

––––––––––––––––––––––––––––––

---

## Referee Comment (RC2) · Anonymous Referee #2 · 25 May 2020

Using idealised 2D ice sheet simulations this paper assess the influence of 2 parameters of the ice constitutive relation that is used in most ice flow models, the Glen flow law. The parameters are the activation energy, Q, and the Glen exponent, n, the pre-exponential factor is adapted to insure consistency between different values of Q and n.

I have several major comments.

First the experimental design is really poorly described and I found very difficult to understand what is really done ; e.g. :
- For the equilibrium state we don't know how the accumulation rate is « adapted » to keep the volume of the ice sheet close to the reference. How close ?
- We understand only in the results section that there is no melting only accumulation.
- Results are presented for several durations (100 years, 2000 years and 10 000 years)

It would be very beneficial to have a clear description of the set-up and experimental design.

Second, as already noted by the first reviewer, the introduction ignores the contributions of several groups to the understanding of the ice rheology. We understand that a review paper by the same first author has been submitted ; however, as this paper is not yet published, the authors should give a better review of previous works to motivate their contribution.

Finally, the flow law that is used here, is certainly the relationship that is used in most ice flow models but I don't think that we can say that it « has so far been assumed certain ». Many models have tested the sensitivity of the models, in different context, to some aspects of the flow law, the enhancement factor (e.g. Ritz et al. 1996, Quiquet, et al. 2018), the initial thermal regime (e.g. Seroussi et al. 2013), the initial viscosity (e.g. Nias et al. 2016, Humbert et al. 2005), the flow law (e.g. Peltier et al., 2000, Pettit and Waddington, 2003, Ma et al., 2010). And there is also a many applications and papers describing anisotropic ice flow models. It is true that this aspect has not been too much explored or discussed in the last community efforts to assess the contribution of the ice sheet to sea level rise. However, the idealised experiment presented here is very simple and it is not clear on which relevant time scales this uncertainty should be taken into account. The results presented in this paper show ice losses of 300 Gt in 1000 years ; this value is the order of magnitude of what is lost by the Greenland ice sheet in one year ; So should we really take into account this uncertainty in sea level rise projections ? Seroussi et al. (2013) have shown that 100 years simulations of the Greenland ice sheet are weekly sensitive to small changes in the initial thermal regime compared to other sources of uncertainty. On the other hand, for longer time scales, the sensitivity of the model results to the ice flow law as already been explored by several authors (see above) for real ice-sheet simulations. In conclusion I found that this simple experiment add very little to existing literature.

In conclusion, I encourage the authors to improve their discussion on previous works to motivate their contribution and run more realistic experiments to better assess the magnitude and time scales relevant for this source of uncertainty.

References :

Humbert, A., Greve, R., and Hutter, K. ( 2005), Parameter sensitivity studies for the ice flow of the Ross Ice Shelf, Antarctica, *J. Geophys. Res.*, 110

Ma, Y., Gagliardini, O., Ritz, C., Gillet-Chaulet, F., Durand, G., Montagnat, M., 2010. Enhancement factors for grounded ice and ice shelves inferred from an anisotropic ice-flow model. Journal of Glaciology 56, 805–812.

Nias I., Cornford, S., and Payne, A. (2016). Contrasting the modelled sensitivity of the Amundsen Sea Embayment ice streams. *Journal of Glaciology, 62*(233), 552-562. doi:10.1017/jog.2016.40

Peltier, W., Goldsby, D., Kohlstedt, D., & Tarasov, L. (2000). Ice-age ice-sheet rheology: Constraints from the Last Glacial Maximum form of the Laurentide ice sheet. *Annals of Glaciology, 30*, 163-176. doi:10.3189/172756400781820859

Pettit, E., and Waddington, E. (2003). Ice flow at low deviatoric stress. *Journal of Glaciology, 49*(166), 359-369. doi:10.3189/172756503781830584

Quiquet, Aurélien et al. "The GRISLI ice sheet model (version 2.0): calibration and validation for multi-millennial changes of the Antarctic ice sheet." Geoscientific Model Development 11 (2018).

Ritz, C., Fabre, A. & Letréguilly, A. Sensitivity of a Greenland ice sheet model to ice flow and ablation parameters: consequences for the evolution through the last climatic cycle. Climate Dynamics **13,** 11–23 (1996).

Seroussi, H., Morlighem, M., Rignot, E., Khazendar, A., Larour, E., & Mouginot, J. (2013). Dependence of century-scale projections of the Greenland ice sheet on its thermal regime. *Journal of Glaciology, 59*(218), 1024-1034. doi:10.3189/2013JoG13J054

---

## Author Comment (AC1) · 24 Jul 2020

**Response to Reviewer Comments**

Journal: The Cryosphere
Title: **Sensitivity of ice flow to uncertainty in flow law parameters in an idealized one-dimensional geometry**

Author(s): Maria Zeitz, Anders Levermann, Ricarda Winkelmann
MS No.: tc-2020-79
MS Type: Research Article

First of all, we would like to thank the editor Alexander Robinson and the two anonymous reviewers for their helpful and excellent comments and their efforts to create the detailed reviews! In our revision of the manuscript we addressed three main issues:

1. We extended the introduction and the discussion sections, based on the reviewers' suggestions and additional literature to give a more comprehensive overview of the state of the art and to allow for an in-depth discussion of our findings.
2. We improved the figures to make them generally more comprehensible and added further results: In particular, we included the discussion of the equilibrium state of the ice sheet before warming in the main part of the paper (see Figs. 2 - 5) and added comparisons to analytically approximated equilibrium states.
3. We further included a new section to investigate how robust our results are when including the effects of surface mass balance changes and sliding (see Figs. 11 and 12).

We provide detailed answers to all comments below. The reviewers' comments are given in black and the authors' in blue.
The changes made to the main document can be found at the end of this document (created with latexdiff).

**Anonymous Referee #1**

This paper presents the results of a very creative and important inquiry into the question of how uncertainty in ice flow-law parameters may impact the predictions numerical ice-sheet models make for future sea level rise. The results of the work are very convincing and the case is well made to attend to flow-law uncertainty with greater effort in the future. The work is conducted with a simple numerical model under simple idealized experiments. Thus, the work reaches substantial conclusions that are not impacted by other, extraneous details. I have put most of my minor editorial comments and questions in the marked-up pdf manuscript that I provide as an attachment to the review. I think that panel B of figure 1 could be re-drafted either using a log scale (not sure if that would work) or just a focus on the top of the ice sheet, so that all the curves don't simply plot one on top of another (as is the case now).

We would like to thank Anonymous Referee #1 very much for their positive evaluation of our manuscript and appreciate their helpful comments.

Please also note the supplement to this comment:
https://www.the-cryosphere-discuss.net/tc-2020-79/tc-2020-79-RC1-supplement.pdf

- Line 1: editorial comment

  done

- Line 7: editorial comment

  done

- Line 9: editorial comment

  done

- Line 20: editorial comment

  done

- Line 24-28: suggestion to cut out the paragraph

  done

- Line 29: editorial comment

  done

- Line 31: "you should probably also cite a paper by Duval and the French group at LGGE, also there are papers by Jacka and Budd"

  Thanks for suggesting this important additional literature! We have now included  the studies by Duva et al. (2010) and Budd and Jacka (1989).

- Line 38: two editorial comments

- Line 40: "I believe lab studies in Grenoble have also found this... The Duval group...."

  Thank you for pointing us to this literature, which we have included in the revised manuscript (in particular, Schulson and Duval (2009) and Duval et al. (2010)). However, we did not find a study by the Duval group which supports the higher value of the flow exponent (n=4). On the contrary, we found literature supporting that the flow exponent might be smaller than the standard value (n<3) (Schulson and Duval (2009) and Duval et al. (2010)).

- Line 53: editorial comment

  done

- Line 72-73: editorial comment

  done

- Line 89-90: editorial comment

  done

- Line 93: editorial comment

  done

- Figure 1: "would a logarithm scale be more useful in separating the profiles here? alternatively, just show the profile at the top of the ice sheet to highlight differences..."

  This is indeed a good point. We added an additional panel in the figure (Fig. 4 in the revised manuscript) to make it clearer how large the differences in initial geometries are.

- Line 148 and 155: missing reference

  done

Anonymous Referee #2

Using idealised 2D ice sheet simulations this paper assess the influence of 2 parameters of the ice constitutive relation that is used in most ice flow models, the Glen flow law. The parameters are the activation energy, Q, and the Glen exponent, n, the pre-exponential factor is adapted to insure consistency between different values of Q and n.

We thank Anonymous Referee #2 very much for their very helpful comments and suggestions for additional literature which will greatly improve the manuscript.

I have several major comments.

First the experimental design is really poorly described and I found very difficult to understand what is really done; e.g.:

- For the equilibrium state we don't know how the accumulation rate is « adapted » to keep the volume of the ice sheet close to the reference. How close?

  Thank you for pointing this out! We have clarified our approach in the revised document.

  All simulations were started with an ice slab of 3 km thickness as an initial state. A first set of equilibrium runs (5 kyrs) was performed, with different parameters n and Q but fixed accumulation rate $a$ of 0.5 m/yr. The ice volume of the steady state depends on the flow parameters, with relative differences between +10% and -15%, compared to the case with standard parameters (see the cross section and the relative differences in volume in Figure 2 of the revised manuscript).

  In order to keep the initial volume similar for all parameter combinations, we therefore increased the accumulation rate for those parameter combinations, which had a lower equilibrium volume and decreased the accumulation rates for the parameter combinations with a higher equilibrium volume. We aimed for less than 1% deviation in total ice volume and repeated simulations with different accumulation rates, until the volumes matched (similar to the bisection method), since an analytical solution cannot be derived for a polythermal ice sheet. The resulting equilibrium states with adapted accumulation rates used for the further analyses deviate by less than 0.8% in ice volume (Supplemental Figure S2).

  We restructured and expanded section 3.1.2, which now describes this process more clearly.

- We understand only in the results section that there is no melting only accumulation.

  Thank you for bringing this to our attention. We hope this will be clearer in the revised manuscript where we adapted the abstract and added two sentences to section 2.5, which state that the climatic mass balance remains unchanged in the warming simulations.

- Results are presented for several durations (100 years, 2000 years and 10 000 years)

  We apologize for the confusion around the timescales: We performed one set of warming simulations for 15.000 years. We here show a close up of the time series of the warming simulations for the first 2.000 and first 100 years as well as timeslices of the relative differences in ice volume and velocities after 100, 1.000 and 10.000 years, as we found that all simulations have reached equilibrium after 10 ka. We added clarifications in section 2 as well as the full time series in Supplemental Figure 3.

It would be very beneficial to have a clear description of the set-up and experimental design.

Second, as already noted by the first reviewer, the introduction ignores the contributions of several groups to the understanding of the ice rheology. We understand that a review paper by the same first author has been submitted; however, as this paper is not yet published, the authors should give a better review of previous works to motivate their contribution.

We are very grateful for the additional literature suggestions and expanded the introduction to include this previous work (see also answers below). In addition, we included the literature basis on which our assessment of the range of the flow parameters n and Q is based.

Finally, the flow law that is used here, is certainly the relationship that is used in most ice flow models but I don't think that we can say that it « has so far been assumed certain ». Many models have tested the sensitivity of the models, in different context, to some aspects of the flow law, the enhancement factor (e.g. Ritz et al. 1996, Quiquet, et al. 2018), the initial thermal regime (e.g. Seroussi et al. 2013), the initial viscosity (e.g. Nias et al. 2016, Humbert et al. 2005), the flow law (e.g. Peltier et al., 2000, Pettit and Waddington, 2003, Ma et al., 2010).

Thank you for bringing this to our closer attention. We agree that the phrase "has so far been assumed certain" is too strong and have revised this in the abstract as well as the main text of the manuscript.

Further, we now discuss the literature concerning the enhancement factor, the initial thermal regime and the mathematical form of the flow law itself in a more detailed way in the manuscript. However, as far as we know there has been no contribution discussing the impact of activation energies Q in the Arrhenius factor in the ice-sheet modeling community, nor has the impact of the flow exponent n been discussed in such a systematic manner.

Generally, changes in the enhancement factor and changes in activation energy Q or flow exponent n are not equivalent, as we now point out more clearly in the introduction and in the discussion: The response to a change in the enhancement factor can be expected to be independent of temperature or stress within the ice, while a change in activation energies Q impacts how the ice sheet responds to changes in the temperature distribution within the ice, and a change in the flow exponent n impacts how the ice sheet responds to changes in stress. Both enhancement factors as well as Q and n affect the ice viscosity, which in PISM is not a result of an optimization procedure but is inferred directly from the system variables, notably the stress and the temperature, and from the water content within the ice.

And there is also a many applications and papers describing anisotropic ice flow models. It is true that this aspect has not been too much explored or discussed in the last community efforts to assess the contribution of the ice sheet to sea level rise.

Indeed, several approaches on how to account for anisotropic ice flow in large ice-sheet models have been put forward by the community. We now discuss those approaches (e.g. Ma et al. 2010) in the introduction of the manuscript. However, as the reviewer also points out, to our knowledge the anisotropic ice-sheet models have not been included in the latest ice-sheet model intercomparison efforts.

It would be an interesting idea for future work to study if the uncertainty in ice flow parameters affects anisotropic ice flow in a similar way as isotropic ice flow. We intentionally limited this study to the isotropic Glen's flow law since it is the most widely used in the ice-sheet modeling community and depends on effectively only four parameters (the flow exponent n, the activation energies for warm and cold ice Q, and the prefactor A, and possibly an enhancement factor which reflects the effect of the anisotropy of the ice). As all of those parameters are less certain than often assumed, we first wanted to scrutinize how these uncertainties play out in the idealized flow-line case.

However, the idealised experiment presented here is very simple and it is not clear on which relevant time scales this uncertainty should be taken into account.

We chose the idealized experiment on purpose, to be able to disentangle the effect of the flow law from other effects like changes in the surface mass balance, but also geothermal heat flux, the topography of the bedrock and the model resolution.

The results indicate that the relative spread in flow-driven ice loss after warming is largest in the first centuries and relatively moderate when approaching equilibrium. A change in flow parameters seems not only to affect the equilibrium shape of the ice sheet itself, but also how fast the ice-sheet reaches the new equilibrium after a change in temperature. We rephrased the results section to clarify the relevance of the different time scales.

The results presented in this paper show ice losses of 300 Gt in 1000 years ; this value is the order of magnitude of what is lost by the Greenland ice sheet in one year ; So should we really take into account this uncertainty in sea level rise projections ?

While this is true, the initial volume of our idealized experiment is approximately 300 times smaller than the volume of the Greenland ice sheet. If the ice losses in our experiment would be scaled up to match the initial volume of the Greenland Ice Sheet, we would expect between 6000 Gt and 30000 Gt of ice loss after 100 years for 6 K of temperature increase, or between 60 and 300 Gt /yr. Further, in contrast to more realistic ice sheet simulations, here, the increased ice flow is the single driver of mass loss.

Relatively, the rates of mass loss in the idealized setup are of a similar order of magnitude compared to mass losses from e.g. the Greenland Ice Sheet (see also section 3.2).

To avoid confusion, we changed the absolute numbers of ice loss to relative numbers compared to the initial volume of the flow-line ice sheet in the revised manuscript. Even though the numbers here might seem small (up to 1% of the total ice volume is lost after 100 years), they compare to observations from the Greenland Ice Sheet, which has lost approximately 0.18% of its total mass in the period between 1972 and 2018 through melting, sliding and flow (assuming a mass loss of 4900Gt (see Mouginot et al., 2019) and a total present day mass of 268,500,000Gt).

Overall, while large uncertainties with respect to the Greenland Ice Sheet mass loss are of course connected to uncertainties in the climate and the melting parameterization (see, e.g., Aschwanden et al. (2019) and ISMIP6), we believe that our study provides an argument that uncertainties in the flow law need to be equally taken into account. We pick this point up in the discussion section of the revised manuscript.

Seroussi et al. (2013) have shown that 100 years simulations of the Greenland ice sheet are weekly sensitive to small changes in the initial thermal regime compared to other sources of uncertainty. On the other hand, for longer time scales, the sensitivity of the model results to the ice flow law as already been explored by several authors (see above) for real ice-sheet simulations. In conclusion I found that this simple experiment add very little to existing literature.

Many thanks for pointing us to these papers, which have indeed provided major contributions to the general understanding of ice-sheet dynamics and future projections. However, while they explore short term and long term effects of variations in rheology on ice loss from realistic ice sheets, the influence of the flow-law parameters themselves on ice velocities and mass balance cannot be inferred from these simulations.

What is new in our manuscript in comparison to previous studies? First, most other studies focus on the influence of enhancement factors, which are linear enhancements of the deformation rate, independent of temperature or effective stress. In contrast, our study specifically focuses on the activation energies $Q$ and the flow exponent $n$. The activation energies determine how the ice softness changes with ice temperature,

which adds a new perspective in particular in the context of a changing climate. The influence of those parameters has not yet been systematically studied as far as we know.

Moreover, the idealized experimental design allows us to disentangle the effect from the flow law parameters from other influences, as e.g. bedrock topography, conditions at the ground, variations in the climatic mass balance etc. It also allows for comparing the simulation results with analytical approximations and thus to understand more deeply the effects of these parameters (which we have now done in sections 3.1 and 3.2).

In conclusion, I encourage the authors to improve their discussion on previous works to motivate their contribution and run more realistic experiments to better assess the magnitude and time scales relevant for this source of uncertainty.

Thank you very much for these suggestions. We extended the discussion on previous work and added a more detailed discussion on the magnitude of the flow parameter induced uncertainty, and the relevant time scales.

We took advantage of the idealized experimental design and complemented our work by analytical considerations (see new results in Figures 3, 5). In addition we estimated how relevant the influence of the flow parameters is when taking other drivers (such as surface mass balance changes or changes in sliding) into account. Including the effects of a lowering surface mass balance leads to similar results with respect to the influence of the flow law parameters (for a reduction of the SMB by 50%, we find a change in relative mass loss from 118% to 190% after 100 years, see Figure 11 in the revised manuscript). We find that with sliding, the effect of the flow uncertainty is even larger (up to +450% after 100 years, see Figure 12).

Encouraged by the referees comments, we improved the discussion about the implications and limitations of our study in section 4.

[revised manuscript text omitted]

---

## Author Response (AR2)

**Response to Editor Comments**

Journal: The Cryosphere
Title: **Sensitivity of ice flow to uncertainty in flow law parameters in an idealized one-dimensional geometry**

Author(s): Maria Zeitz, Anders Levermann, Ricarda Winkelmann
MS No.: tc-2020-79
MS Type: Research Article

We would like to thank the editor Alexander Robinson for his helpful comments! We provide detailed answers to all comments below. In particular, we added a new figure, which addresses the main question brought up by the editor, to the supplemental material. We are just as happy to move this figure to the main document if you see fit.
The editor's comments are given in black and the authors' in blue. The changes made to the manuscript can be found at the end of this document (created with latexdiff).

**Editor Decision: Publish subject to minor revisions (review by editor)** (16 Aug 2020) by Alexander Robinson
Comments to the Author:

The manuscript is in very good condition, and near-ready for publication, except for potentially one critical point that I believe has been overlooked until now.

I apologize that I didn't notice it until now, but I am somewhat concerned by the experimental setup described in Section 3.2, in which the initial volume is kept constant by tuning accumulation. This procedure may introduce an unwanted change in the temperature of the ice sheet, since the accumulation rate directly controls the vertical velocity profile, which enters the thermodynamic equation. By e.g., increasing accumulation, one can generally expect a colder ice sheet and a lower rate factor. Accumulation rate differences of between 50% and 300% seem large enough to have a strong impact on internal temperature. Since flow differences due to Q and n are precisely what is studied here, I am worried that the interpretation of the results is contaminated by the systematic changes in accumulation.

The best solution at this stage would be to be able to confirm that this is not the case – i.e., that despite the influence of accumulation on internal temperature, its impact is much smaller than that of the rate factor parameters themselves. Maybe this could be done with a fixed ice thickness profile: then you could diagnose changes in the rate factor and/or ice velocity as a function of the rate-factor parameters and accumulation and see the magnitude of their impacts? If changes in accumulation do play a large role, perhaps you could consider simply allowing the initial volumes to differ – since you report volume losses in percent, the difference in absolute volume should not have a large impact given the idealized configuration of the experiment.

Thank you very much for bringing this up. In order to keep results most comparable we chose to fix the equilibrium volume and the geometry of the ice sheet though adapted accumulation rates. Indeed, the temperature distribution in the ice changes with variations in the accumulation rate, as we show in

Supplementary Figure S4 (a) of the revised supplementary material. Temperatures decrease with increased accumulation rates, which in this setup are associated with high activation energies, and vice versa. The temperature change due to adapted accumulation rates is most prominent at the top of the ice sheet.

The softness (or rate factor) of the ice depends on pressure adjusted temperature and flow parameters. In this setup high activation energies are associated with high softness and increased accumulation rates and low activation energies with low softness and low accumulation rates. The temperature decrease from the increased accumulation rates thus counteracts the increase in softness due to the change in parameters.

The equilibrium simulations with adapted accumulation rates show distinct variations in softness, in particular close to the base of the ice sheet (see Figure S4 b). We disentangle the influences of the change in flow parameters (i.e. activation energies) and the change in the temperature distribution: Replacing the flow parameters by the standard values leads to very similar softness profiles, despite changes in the ice-sheet temperature (see S4 c). On the other hand, fixing the ice temperatures and varying only the flow parameters reproduces the softness profiles which are observed in equilibrium (see S4 d).

We included a short discussion with a reference to the supplementary figure in the main text.

In either case, I think it is important to address this point before moving forward with publication.

Please find additional minor points below:

L42: I suggest separating discussion and references about the parameter n and the parameter Q. Therefore, I would move the sentences of L52-56 concerning the parameter n to just after "which can be between 2 and 4" on L42, to keep the discussion of n together. Then proceed with discussion of Q. Ie, "..., which can be between 2 and 4 [REFS]. [Sentences from L52-56]. The activation energies Q ... can also vary by a factor of two [REFS].
done

L43: 50+ citations to support the statement that n shows a large spread and that the activation energies "can vary by a factor of two" seems a bit excessive. I support the notion of such an extensive review of older and perhaps now forgotten literature, especially if nothing new has been done since beyond it. However, for that I would expect some specific information that came from each paper with separated citations (essentially, a more detailed review paragraph). If you are only interested in supporting the above statement, I would suggest reducing the number of references to the most relevant and ground-breaking ones.
done

L52: Merge "(Zeitz et al. submitted)" into the other citations here, or specify what is different about the reference, aside from only having been submitted. As it is now, it's not clear why it stands apart.

L75: It may depend, among others, on => It may depend on water content ..., among other things.
done

L79: temperature difference => temperature relative
done

L95: independently => independent
done

L115: the typical standard value => a typical reference value
done

L116: the standard value => a typical reference value
done

L118: The standard values serve as a reference point and correspond => The reference values above correspond
done

L130: warm or cold ice respectively => cold or warm ice, respectively,
done

L134: in the same order => on the same order
done

L146: I would expect to see the basal boundary condition here as well, ie, geothermal heat flux.
done

L196: the volume calculated => the ice profile
done

L210: , which mathces => that matches
done

Fig. 6: Remove grey lines connecting the zoom of panel (b) to that of panel (a). They do not help comprehension, and the caption and axes are clear enough.
done

Fig. 8: Panel (b) should include a y-axis label as well (despite having the same units, I assume), to avoid ambiguity.
done

L259-261: I'm not able to follow this logic – it sounds like something contradictory is stated. If $A_0$ reduces with a lower n, or rather if $A_0$ increases with a higher n, then the rate factor should increase with a higher n. This would lead to faster flow too. Perhaps another sentence or two would help with clarity of the message here. Or simply delete the sentence "This might be linked …". The last sentence of the paragraph is clear and seems warranted by the results.
done

L270: shallow ice approximation => shallow shelf approximation [?]
done

L290: allows to => allows us to
done

L296: allows to => allows us to
done

L328: by e.g => by e.g.
done

L330: to assess => to assessing
done

L331: allows informed => helps to inform
done

L332: the Greenland or Antarctic Ice Sheet => continental-scale ice sheets [since these insights could apply to any ice sheet, right?]
done

[revised manuscript text omitted]

**Figure S4. Effect of differences in temperature and different activation energies on ice softness in equilibrium: (a)** Pressure adjusted temperature of the ice sheet in the equilibrium state for different combinations of activation energies $Q_c, Q_w$ (see title of each tile) with a flow exponent $n = 3$. Note that the accumulation rates are adjusted to keep the equilibrium volume fixed, with decreased accumulation rates for low activation energies and increased accumulation rates for high accumulation energies. The increase in accumulation leads to cooler temperatures at the top of the ice sheet. **(b)** Ice softness $A$ in equilibrium for different combinations of activation energies $Q_c$ and $Q_w$. A clear increase in softness, in particular at the base of the ice sheet is observed for high activation $Q_c$ and $Q_w$. **(c)** Ice softness calculated from the pressure adjusted temperatures shown in **(a)** but fixing $Q_w$ and $Q_c$ at the reference values. The changes in temperature distribution alone are not sufficient to reproduce the softness shown in panel **(b)**. **(d)** Ice softness calculated from a fixed pressure adjusted ice temperature and different $Q_c$ and $Q_w$ parameters. The change in activation energies alone can reproduce the softness shown in panel **(b)**. Here we show equilibrium values without a warming anomaly. A very similar behavior is observed for the equilibrium states with increased temperatures.

---

## Author Response (AR3)

**Response to Editor Comments**

Journal: The Cryosphere

Title: **Sensitivity of ice flow to uncertainty in flow law parameters in an idealized one-dimensional geometry**

Author(s): Maria Zeitz, Anders Levermann, Ricarda Winkelmann
MS No.: tc-2020-79
MS Type: Research Article

Dear Alex Robinson,

Thank you very much for spotting the inconsistency! Indeed, a constant heat flux of $42mW/m^2$ was prescribed. We are sorry for the confusion and now corrected the manuscript. The information is now found in line 138 of the corrected manuscript.

Sincerely,
Maria Zeitz, Anders Levermann, and Ricarda Winkelmann